# A Design Space Study for LISTA and Beyond

**Tianjian Meng**[1]*, **Xiaohan Chen**[2]*, **Yifan Jiang**[2], **Zhangyang Wang**[2]
[1]Google Research   [2]The University of Texas at Austin
mengtianjian@google.com
{xiaohan.chen, yifanjiang97, atlaswang}@utexas.edu

## Abstract

In recent years, great success has been witnessed in building problem-specific deep networks from unrolling iterative algorithms, for solving inverse problems and beyond. Unrolling is believed to incorporate the model-based prior with the learning capacity of deep learning. This paper revisits *the role of unrolling as a design approach for deep networks*: to what extent its resulting special architecture is superior, and can we find better? Using LISTA for sparse recovery as a representative example, we conduct the first thorough *design space study* for the unrolled models. Among all possible variations, we focus on extensively varying the connectivity patterns and neuron types, leading to a gigantic design space arising from LISTA. To efficiently explore this space and identify top performers, we leverage the emerging tool of neural architecture search (NAS). We carefully examine the searched top architectures in a number of settings, and are able to discover networks that are consistently better than LISTA. We further present more visualization and analysis to "open the black box", and find that the searched top architectures demonstrate highly consistent and potentially transferable patterns. We hope our study to spark more reflections and explorations on how to better mingle model-based optimization prior and data-driven learning.

## 1 Introduction

The signal processing and optimization realm has an everlasting research enthusiasm on addressing ill-conditioned inverse problems, that are often regularized by handcrafted model-based priors, such as sparse coding, low-rank matrix fitting and conditional random fields. Since closed-form solutions are typically unavailable for those model-based optimizations, many analytical iterative solvers arise to popularity. More recently, deep learning based approaches provide an interesting alternative to inverse problems. A learning-based inverse problem solver attempts to approximate the inverse mapping directly by optimizing network parameters, by fitting "black box" regression from observed measurements to underlying signals, using synthetic or real-world sample pairs.

Being model-based and model-free respectively, the analytical iterative solvers and the learning-based regression make two extremes across the spectrum of inverse problem solutions. A promising direction arising in-between them is called *algorithm unrolling* (Monga et al., 2019). Starting from an analytical iterative solver designed for model-based optimization, its unrolled network architecture can be generated by cascading the iteration steps for a finite number of times, or equivalently, by running the iterative algorithm with early stopping. The original algorithm parameters will also turn into network parameters. Those parameters are then trained from end to end using standard deep network training, rather than being derived analytically or selected from cross-validation.

Unrolling was first proposed to yield faster trainable regressors for approximating iterative sparse solvers (Gregor & LeCun, 2010), when one needs to solve sparse inverse problems on similar data repeatedly. Later on, the unrolled architectures were believed to incorporate model-based priors while enjoying the learning capacity of deep networks empowered by training data, and therefore became a rising direction in designing principled and physics-informed deep architectures. The growing popularity of unrolling lies in their demonstrated effectiveness in developing compact, data-efficient, interpretable and high-performance architectures, when the underlying optimization model is assumed available. Such approaches have witnessed prevailing success in applications such as compressive

---

*The authors Tianjian Meng and Xiaohan Chen contribute equally to the work.

sensing (Zhang & Ghanem, 2018), computational imaging (Mardani et al., 2018), wireless communication (Cowen et al., 2019; Balatsoukas-Stimming & Studer, 2019), computer vision (Zheng et al., 2015; Peng et al., 2018), and other algorithms such as ADMM (Xie et al., 2019).

The empirical success of unrolling has sparkled many curiosities towards its deeper understanding. A series of efforts (Moreau & Bruna, 2017; Giryes et al., 2018; Chen et al., 2018; Liu et al., 2019; Ablin et al., 2019; Takabe & Wadayama, 2020) explored the theoretical underpinning of *unrolling as a specially adapted iterative optimizer* to minimizing the specific objective function, and proved the favorable convergence rates achieved over classical iterative solver, when the unrolled architectures are trained to (over)fit particular data. Orthogonally, this paper reflects on *unrolling as a design approach for deep networks*. The core question we ask is:

> *For solving model-based inverse problems, what is the role of unrolling in designing deep architectures? What can we learn from unrolling, and how to go beyond?*

## 1.1 RELATED WORKS: PRACTICES AND THEORIES OF UNROLLING

(Gregor & LeCun, 2010) pioneered to develop a learning-based model for solving spare coding, by unrolling the iterative shrinkage thresholding algorithm (ISTA) algorithm (Blumensath & Davies, 2008) as a recurrent neural network (RNN). The unrolled network, called Learned ISTA (LISTA), treated the ISTA algorithm parameters as learnable and varied by iteration. These were then fine-tuned to obtain optimal performance on the data for a small number of iterations. Numerous works (Sprechmann et al., 2015; Wang et al., 2016a; Zhang & Ghanem, 2018; Zhou et al., 2018) followed this idea to unroll various iterative algorithms for sparse, low-rank, or other regularized models.

As the most classical unrolling example, a line of recent efforts have been made towards theoretically understanding LISTA. Moreau & Bruna (2017) re-factorized the Gram matrix of the dictionary, and thus re-parameterized LISTA to show its acceleration gain, but still sublinearly. Giryes et al. (2018) interpreted LISTA as a projected gradient descent descent (PGD) where the projection step was inaccurate. Chen et al. (2018) and Liu et al. (2019) for the first time introduced necessary conditions for LISTA to converge linearly, and show the faster asymptotic rate can be achieved with only minimal learnable parameters (e.g., iteration-wise thresholds and step sizes). Ablin et al. (2019) further proved that learning only step sizes improves the LISTA convergence rate by leveraging the sparsity of the iterate. Besides, several other works examined the theoretical properties of unrolling other iterative algorithms, such as iterative hard thresholding (Xin et al., 2016), approximated message passing (Borgerding & Schniter, 2016) , and linearized ADMM (Xie et al., 2019). Besides, the safeguarding mechanism was also introduced for guiding learned updates to ensure convergence, even when the test problem shifts from the training distribution (Heaton et al., 2020).

On a separate note, many empirical works (Wang et al., 2016a;b; Gong et al., 2020) advocated that the unrolled architecture, when used as a building block for an end-to-end deep model, implicitly enforces some structural prior towards the model training (resulting from the original optimization objective) (Dittmer et al., 2019). That could be viewed as a special example of "architecture as prior" (Ulyanov et al., 2018). A recent survey (Monga et al., 2019) presents a comprehensive discussion. Specifically, the authors suggested that since iterative algorithms are grounded on domain-specific formulations, they embed reasonably accurate characterization of the target function. The unrolled networks, by expanding the learnable capacity of iterative algorithms, become "tunable" to approximate the target function more accurately. Meanwhile compared to generic networks, they span a relatively small subset in the function space and therefore can be trained more data-efficiently.

## 1.2 MOTIVATIONS AND CONTRIBUTIONS

This paper aims to **quantitatively** assess "how good the unrolled architectures actually are", using LISTA for sparse recovery as a representative example. We present the first **design space ablation study**[1] on LISTA: starting from the original unrolled architecture, we extensively vary the connectivity patterns and neuron types. We seek and assess good architectures in a number of challenging settings, and hope to expose successful design patterns from those top performers.

As we enable layer-wise different skip connections and neuron types, the LISTA-oriented design space is dauntingly large (see Sections 2.1 and 2.2 for explanations). As its manual exploration is infeasible, we introduce the tool of neural architecture search (NAS) into the unrolling field. NAS

---

[1]We define a *design space* as a family of models derived from the same set of architecture varying rules.

can explore a gigantic design space much more efficiently, and can quickly identify the subset of top-performing candidates for which we can focus our analysis on. Our intention is **not** to innovate on NAS, but instead, to answer a novel question in LISTA using NAS as an experiment tool.

From our experiments, a befitting quote may be: for designing neural networks to solve model-based inverse problems, *unrolling is a good answer, but usually not the best.* Indeed, we are able to discover consistently better networks in all explored settings. From top candidates models, we observe highly **consistent** and potentially **transferable** patterns. We conclude this paper with more discussions on how to better leverage and further advance the unrolling field.

## 2 TECHNICAL APPROACH

Assume a sparse vector $\boldsymbol{x}^* = [x_1^*, \cdots, x_M^*]^T \in \mathbb{R}^M$, we observe its noisy linear measurements:

$$\mathbf{b} = \sum\nolimits_{m=1}^M \mathbf{d}_m x_m^* + \varepsilon = \mathbf{D}\mathbf{x}^* + \varepsilon, \tag{1}$$

where $\mathbf{b} \in \mathbb{R}^N$, $\mathbf{D} = [\mathbf{d}_1, \cdots, \mathbf{d}_M] \in \mathbb{R}^{N \times M}$ is the *dictionary*, and $\varepsilon \in \mathbb{R}^N$ is additive Gaussian white noise. For simplicity, each column of $\mathbf{D}$ is normalized, that is, $\|\mathbf{d}_m\|_2 = \|\mathbf{D}_{:,m}\|_2 = 1$, $m = 1, 2, \cdots, M$. Typically, we have $N \ll M$. A popular approach for solving the inverse problem: $\mathbf{b} \to \mathbf{x}$, is to solve the LASSO below ($\lambda$ is a scalar):

$$\min_{\mathbf{x}} \frac{1}{2}\|\mathbf{b} - \mathbf{D}\mathbf{x}\|_2^2 + \lambda\|\mathbf{x}\|_1 \tag{2}$$

using iterative algorithms such as the iterative shrinkage thresholding algorithm (ISTA):

$$\mathbf{x}^{(k+1)} = \eta_{\lambda/L}\left(\mathbf{x}^{(k)} + \mathbf{D}^T(\mathbf{b} - \mathbf{D}\mathbf{x}^{(k)})/L\right), \quad k = 0, 1, 2, \ldots \tag{3}$$

where $\eta_\theta$ is the soft-thresholding function[2] and $L$ is a smoothness constant that decides the step size.

Inspired by ISTA, (Gregor & LeCun, 2010) proposed to learn the weight matrices in ISTA rather than fixing them. LISTA unrolls ISTA iterations as a recurrent neural network (RNN): if truncated to $K$ iterations, LISTA becomes a $K$-layer feed-forward neural network with side connections:

$$\mathbf{x}^{(k+1)} = \eta_{\theta^{(k)}}(\mathbf{W}_\mathbf{b}\mathbf{b} + \alpha^{(k)}\mathbf{W}_\mathbf{x}^{(k)}\mathbf{x}^{(k)}), \quad k = 0, 1, \cdots, K-1. \tag{4}$$

If we set $\mathbf{W}_\mathbf{b} \equiv \mathbf{D}^T/L$, $\mathbf{W}_\mathbf{x}^{(k)} \equiv \mathbf{I} - \mathbf{D}^T\mathbf{D}/L$, $\alpha^{(k)} \equiv 1$, $\theta^{(k)} \equiv \lambda/L$, then LISTA recovers ISTA[3]. In practice, we start with $\mathbf{x}^{(0)}$ set to zero. As suggested by the seminal work (Liu et al., 2019), sharing $\mathbf{W}_\mathbf{b}$ and $\mathbf{W}_\mathbf{x}$ across layers does no hurt to the performance while reducing the parameter complexity. We follow their *weight-tying* scheme in the paper, while learning layer-wise $\alpha^{(k)}$ and $\theta^{(k)}$. $\alpha^{(k)}$ is separated from $\mathbf{W}_\mathbf{x}$ to preserve flexibility.

We note one slight difference between our formulation (4), and the parameterization scheme suggested by Theorem 1 of (Chen et al., 2018). The latter also restricted $\{\mathbf{W}_\mathbf{b}, \mathbf{W}_\mathbf{x}\}$ to be coupled layerwise, and showed it yielded the same representation capability as the orginal LISTA in Gregor & LeCun (2010). We instead have $\{\mathbf{W}_\mathbf{b}, \mathbf{W}_\mathbf{x}\}$ as two independent learnable weight matrices. The main reason we did not follow (Chen et al., 2018) is that $\mathbf{W}_\mathbf{b}$-$\mathbf{W}_\mathbf{x}$ coupling was directly derived for unrolling ISTA, which is no longer applicable nor intuitive for other non-LISTA architecture variants. Our empirical results also back that our parameterization in (4) is easier to train for varied architectures from search, and performs the same well when adopted in training the original LISTA.

Now, given pairs of sparse vector and its noisy measurement $(\mathbf{x}^*, \mathbf{b})$, our goal is to learn the parameters $\Theta = \{\mathbf{W}_\mathbf{b}, \mathbf{W}_\mathbf{x}, \alpha^{(k)}, \theta^{(k)}\}_{k=0}^{K-1}$ such that $\mathbf{x}^{(K)}$ is close to $\mathbf{x}^*$ for all sparse $\mathbf{x}^*$ following some distribution $\mathcal{P}$. Therefore, all parameters in $\Theta$ are subject to end-to-end learning:

$$\min_\Theta \mathbb{E}_{\mathbf{x}^*, \mathbf{b} \sim \mathcal{P}}\|\mathbf{x}^{(K)}(\Theta, \mathbf{b}, \mathbf{x}^{(0)}) - \mathbf{x}^*\|_2^2. \tag{5}$$

This problem is approximately solved over a training dataset $\{(\mathbf{x}_i^*, \mathbf{b}_i)\}_{i=1}^N$ sampled from $\mathcal{P}$.

---

[2]Soft-thresholding function is defined in a component-wise way: $\eta_\theta(\mathbf{x}) = \text{sign}(\mathbf{x})\max(0, |\mathbf{x}| - \theta)$

[3]Those can be parameter initialization in LISTA, which we follow (Chen et al., 2018) to use by default.

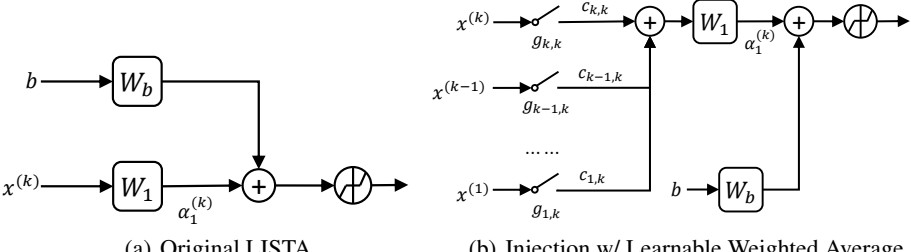

|  (a) Original LISTA  |  (b) Injection w/ Learnable Weighted Average |

Figure 1: An illustration of (a) classic LISTA; (b) connection injection with learnable weighted average. We only plot one layer for illustration.

## 2.1 VARYING THE CONNECTIVITY IN LISTA

We first present how we construct new LISTA-oriented architectures based on extensively varying the connectivity patterns, starting from the original unrolled LISTA. The way we add connections to the original unrolled model is called *Learnable Weighted Average* (LWA). Specifically, before the output of the $k$-th layer $x^{(k)}$ is fed into the next layer, we allow the outputs of previous layers to be possibly (not necessarily) connected to $x^{(k)}$. The input to the next layer is calculated with a learnable average of all the connections. Formally,

$$\tilde{\mathbf{x}}^{(k)} = \sum\nolimits_{i=1}^{k} g_{i,k} \cdot \left( c_{i,k} \, \mathbf{x}^{(i)} \right), \tag{6}$$

where $c_{i,k}$ are (unbounded) model parameters that will be trained with data, and $g_{i,k}$ are gates whose values are chosen to decide specific connectivity. Note that LWA introduces a few extra parameters (coefficients $c_{i,k}$) than LISTA, but those constitute only a very minor portion w.r.t. the size of $\Theta$.

We completely recognize that there are other possibilities to vary connections, and did our due diligence to make the informed choice of LWA. First, we only focus on adding connections, motivated from both the deep learning and the empirical unrolling fields. On one hand, prevailing success in deep learning (He et al., 2016; Huang et al., 2017) endorse that more densely connected networks are more likely to yield better performance. Prior unrolling works (Wang et al., 2016a) also empirically found that removing connections from an unrolled architectures would deteriorate its effectiveness. We also provide additional experiments in the **Appendix** A.4 to demonstrate that pruning existing connections in LISTA will be consistently harmful. Second, besides injecting new connections by LWA, we have also experimented with several other options, including a naive averaging way (simply averaging the outputs of all chosen layers), and a momentum-inspired averaging way (which is inspired by placing a momentum term into the unrolled algorithm). Through various experiments, we have confirmed that both options are almost consistently inferior to LWA, and therefore focus on LWA in the main paper. The details of the two other options, as well as the experimental comparisons, can be founded in the **Appendix** A.3 too.

We by default keep all existing connections in LISTA itself, i.e. the gate function $g_{k,k}$ that controls the connection from the immediate preceding layer is always 1. In this case, for a $K$-layer network, the total number of possible architectures is $2^{1+2+\cdots+(K-2)} = 2^{\frac{(K-1)(K-2)}{2}}$. When $K = 16$ as commonly set in previous unrolling works (Chen et al., 2018; Borgerding & Schniter, 2016), only varying the connnecitivy will lead to a total of over $2^{105} \approx 4 \times 10^{31}$ different architectures.

## 2.2 VARYING THE NEURON TYPE IN LISTA

Besides connection patterns, another important factor that might affect the model performance is the choice of the neuron (non-linear activation function) type. The default neuron in LISTA is the soft-thresholding operator with learnable thresholds (bias term), naturally from unrolling ISTA.

To study whether they might be potential better neuron options or combinations, we augment the search space to allow each layer to select its neuron from three types:{soft-thresholding, ReLU, LeakyReLU($\alpha = 0.1$)}. That increases the total architecture numbers further by $3^{16}$ times, resulting in the final gigantic space of $\sim 1.75 \times 10^{39}$ candidate architectures deriving from a 16-layer LISTA.

## 2.3 HOW TO EFFICIENTLY FIND AND ANALYZE GOOD ARCHITECTURES

The daunting size of our LISTA search space begs the question: how to effectively find the best small subset of architectures, that we can observe and reliably summarize patterns from? A manual

exploration is infeasible, and hence we refer to the tool of neural architecture search (NAS) (Zoph & Le, 2016) to fulfill our goal. Here NAS is leveraged as an off-the-shelf tool to explore the search space efficiently and to quickly focus on the most promising subset of top candidates.

**Are NAS results stable and trustworthy?**   Since our main goal is to analyze the found top-performer models and to identify design patterns, the effectiveness and stability of NAS are crucial to the trustworthiness of our conclusions. We take several actions to address this potential concern.

First, to avoid the instability arising in training and evaluating the sampled candidate models, we adopt no parameter sharing (Pham et al., 2018; Liu et al., 2018). Parameter sharing is currently a popular scheme in NAS to save search time, but can introduce search bias (Chu et al., 2019). Despite the accompanied tremendous computation costs, we train each individual model from scratch to fully converge, to ensure the most stable, thorough and fully reproducible search results as possible.

Second, to avoid the possible artifact of any single NAS algorithm, we repeat our experiments using three different state-of-the-art, non-weight-sharing NAS algorithms, including reinforcement learning (RL) based method (Zoph & Le, 2016), regularized evolution based method (Real et al., 2019) and random sampling based method (Li & Talwalkar, 2019). Those algorithms usually yield diverse search behaviors, and in general there is no clear winner among the three (Li & Talwalkar, 2019). We use the negative validation NMSE as the optimization reward for them all. Somehow surprisingly, we find: (1) the best candidate architectures (e.g., top-50 in terms of NMSE) found by the three methods reach almost the same good NMSE values; and (2) those best subsets are highly overlapped (nearly identical) in the specific models found, and show quite consistent architecture patterns. Hence due to the space limit, we focus on reporting all results and findings from RL-based NAS, as the other two produce almost the same. We hypothesize that our gigantic LISTA search space might have a smoother and more friendly landscape compared to typical NAS benchmarks, that facilitates state-of-the-art NAS algorithms to explore. We leave that for future work to verify.

Third, the top-1 architecture found by NAS will inevitably be subject to sampling randomness and other search algorithm fluctuations (e.g., hyperparameters). Therefore, instead of overly focusing on analyzing the top-1 or few architectures, we take one additional step to create an **"average architecutre"** from the top-50 architectures, to further eliminate random factors that might affect our results' credibility. Specifically, for every possible connection between layers, we compute the percentage (between 0 and 1) of models from the top-50 set that have this connection. We then use a threshold of 0.5 to decide whether the average architecture will have this connection (if the percentage $\geq 0.5$) or not (if $< 0.5$). As we will visualize in Section 4, the top-50 architectures usually have a high agreement level on connections, making most percentages naturally close to either 0 or 1. The average architecture helps more clearly perceive the "good patterns" shared by top models.

## 3   EXPERIMENTS AND ANALYSIS

### 3.1   EXPERIMENT SETTING AND IMPLEMENTATION DETAILS

**Searching.** As discussed in Section 2.3, we focus on the RL-based NAS (Zoph & Le, 2016) without weight sharing, i.e., train sampled models from scratch to full convergence. Our experiments are performed on 400 Cloud TPUs, with more than 25,000 architectures sampled during each search[4].

After the search is complete, we focus on the "average architecture" from the top-50 sampled set, as defined in Section 2.3. Typical NAS works that often find the few best models to have notable performance gaps between each other; different from them, our top-50 searched models have only minor NMSE deviations, in addition to their consistent architecture patterns. That endorses the stability of our search, and potentially reaffirms our smoothness assumption of LISTA search space.

To emphasize, we have only performed NAS in the problem setting of Section 3.2 (synthetic, noiseless). For all remaining experiments, we directly re-train and evaluate the same average architecture, on corresponding training and testing sets. The main motivation is to *avoid the hassle of re-doing the tedious model search for every new problem*. Instead, we wish to show that by searching only once in a problem setting, the searched architecture not only performs best in the same setting, but also can be directly applied (with re-training) to other related settings with superior performance. Such *transferrability* study has also been a popular subject in NAS literature (Zoph et al., 2018), where one searches on one dataset and re-trains the searched model on another similar dataset.

---

[4]Our codes, and searched & trained models are published on GitHub.

Table 1: Summary of all synthetic experiment results: Comparing our searched architecture (from Section 3.2) with three baselines. All performance numbers are in decibel (the lower the better).

| Model | LISTA | LFISTA | Dense-LISTA | Searched |
|---|---|---|---|---|
| [a] Number of extra connections (w.r.t. LISTA) | 0 | 14 | 105 | 42 |
| [b] SNR = $\infty$ (the ideal setting where NAS is performed) | -43.3 | -47.3 | -48.3 | **-54.2** |
| [c] Gaussian Noise: SNR=40 | -34.3 | -35.3 | -37.3 | **-38.2** |
| [d] Gaussian Noise: SNR=20 | -16.5 | -16.9 | -17.8 | **-18.7** |
| [e] Non-exactly sparse $\mathbf{x}^*$ | -22.4 | -26.3 | -26.9 | **-29.1** |
| [f] Transfer-Noise (Gaussian): SNR=$\infty \rightarrow$ SNR=40 | -32.4 | -33.1 | -33.7 | **-34.3** |
| [g] Transfer-Noise (Gaussian): SNR=$\infty \rightarrow$ SNR=20 | -7.6 | -13.0 | -13.3 | **-13.3** |
| [h] Transfer-Noise (Gaussian $\rightarrow$ S&P): density 1% | -6.0 | -9.2 | -11.3 | **-11.7** |
| [i] Perturbed dictionary | -28.9 | -29.8 | -30.4 | **-30.4** |
| [j] Limited Data (Training size 10,240) | -32.4 | -42.6 | -38.4 | **-49.6** |
| [k] Limited Data (Training size 5,120) | -26.0 | -35.3 | -26.9 | **-35.4** |

**Training and evaluating sampled models.** We adopt the same stage-wise training strategy from (Chen et al., 2018; Liu et al., 2019) and use the same default $K = 16$ layers. Every model is trained with batch size 128 and learning rate 0.0005 on 2 Cloud TPUs. In most experiments, the training, validation and testing sets are sampled i.i.d. without overlap, except the training/testing mismatch experiments where we purposely have training and testing sets sampled from different distributions. Also by default, we sample 102,400 samples for training, and 10,240 for validation and testing each, constituting *data abundant* settings. A *data limited* setting will be discussed later. For all synthetic data experiments, the model performance is evaluated with normalized MSE (NMSE) in decibel (Chen et al., 2018) averaged over the testing set: the lower the better.

**Baselines.** We compare with three strong hand-crafted baselines: the original unrolling **LISTA** model; the densely connected variant of LISTA (every layer connected to all its preceding layers), denoted as **Dense-LISTA**; and another improved unrolled sparse recovery model from FISTA (Moreau & Bruna, 2017) called **LFISTA**, which could also perceived as model-based injection of more connections into LISTA. For the fair comparison with LFISTA, we adopt the same parameterization to (4) and (6) to model its Nesterov acceleration term. We find that to consistently improve the training stability and final performance of LFISTA too. More details are in the **Appendix** A.1.

## 3.2 Search and Evaluation in the Noiseless Setting

We perform our search on the simplest synthetic setting with no additive noise, i.e. $\varepsilon = 0$ in (1). We generate all our data, with a random sampled dictionary $D \in \mathbb{R}^{m \times n}$ with $m = 250, n = 500$, following (Chen et al., 2018; Borgerding & Schniter, 2016). In **Appendix** A.2, we also investigate the effect of problem size by varying $m$ and $n$, as well as the effect of an ill-conditioned dictionary (instead of Gaussian). We sample the entries of $D$ i.i.d. from the Gaussian distribution $D_{ij} \sim N(0, 1/m)$ and normalize its columns to unit $\ell_2$ norm. To sample the sparse vectors $x$, we decide each of its entry to be non-zero following the Bernoulli distribution with $p = 0.1$, and then generate its non-zero entries from the $N(0, 1)$. We tried to adjust the dictionary formulation, sparsity level or nonzero magnitude distributions in some experiments, and observe the conclusions to be highly consistent. We therefore only report in this setting due to the space limit.

The results are reported and in the row [b] in Table 1, where we compared our searched top-50 average architecture with three baselines. Our main observations are:

- **The searched average model (significantly) outperforms hand-crafted ones.** That proves our concept: much stronger models could be found by NAS in the LISTA-oriented design space. Particularly, it surpasses the vanilla LISTA by a large gap of 10.9 dB.

- **The choices of neuron types are "embarrassingly uniform".** Even we allow each of the 16 layers to independently select neurons, none shows to favor ReLU or leaky ReLU. All our top-50 architectures unanimously adopt soft-thresholding as the only neuron type for all their layers. It is somehow a surprise, and hints the important role of model-based prior in unrolled networks.

- **Non-trivial connectivity patterns are discovered,** While LFISTA and Dense-LISTA both improve over LISTA, our search result indicates that "the denser the better" is not the true claim here (Table 1, row [a]). That differs from the general mind in computer vision models (Huang et al., 2017). More discussions on the searched connectivity are in Section 4.

### 3.3 TRASNFERABILITY STUDY FOR THE SEARCHED MODEL

We now present the transferablity study, to re-train and evaluate the above-searched average architecture in more challenging settings. The baselines are also re-trained and compared in fair settings.

**#1. Noisy measurement** $[c, d]$    We first add non-zero Gaussian noise $\varepsilon$ (i.i.d. across training and testing) to synthetic data. We use two noise levels, corresponding to SNR = 40 dB and 20 dB, respectively. Results in rows $[c]$, $[d]$ of Table 1 show similar trends as $[a]$, though with small gaps.

**#2. Non-exactly sparse vector** $[e]$    We next challenge LISTA to recover the non-exactly sparse $\mathbf{x}^*$ in (1). We generate $\mathbf{x}^*$ from the Gamma distribution $\Gamma(\alpha, \beta)$ with $\alpha = 1.0$ and $\beta = 0.1$. In this way, $\mathbf{x}^*$ has a handful of coordinates with large magnitudes, while the remaining coordinates have small yet nonzero values, making it approximately sparse or not exactly so. In order to disentangle different factors' influences, no additive noise is placed in this case (SNR = $\infty$). From row $[e]$ of Table 1, the same trend is observed, so is the superiority of the searched model.

**#3. Training-testing mismatch** $[f, g, h, i]$    We proceed to checking more challenging and practical scenarios of model robustness where the training and testing distribution may mismatch. We consider three cases in Table 1: (1) Transfer-Noise (Gaussian): we train models on the noiseless training set (SNR=$\infty$), and directly apply to the two noisy testing sets, i.e., SNR = 40 dB as in row $[f]$ and 20 dB as in $[g]$. (2) Transfer-Noise (Gaussian $\rightarrow$ salt & pepper): the models trained with SNR = $\infty$ are encountered with an unseen noise type in testing: salt & pepper noise with 1% density; (3) Perturbed dictionary[5]: we keep our training/validation sets generated by the same dictionary $D$ (SNR = $\infty$), while re-generating a testing set by the following way: we sample a small $\Delta D$ from a Laplacian distribution $L(0, 10^{-3})$, perturb $\bar{D} = D + \Delta D$ to create an unseen basis, normalize $\bar{D}$ to unit column and then use it to generate new testing samples (no additive noise).

Not surprisingly, all models see degraded performance under those challenging settings, but our searched architecture sticks as the best (or tie) among all. LFISTA and Dense-LISTA are also able to outperform LISTA. Besides, the gaps between Dense-LISTA and ours are generally reduced in all those mismatch cases, implying that denser connections might benefit model robustness here.

**#4. Limited training data** $[j, k]$    We lastly verify a hypothesis raised in (Monga et al., 2019), that unrolling provides model-based prior and helps train more generalizable networks in the data-limited training regime. To prove this concept, we reduce the training set to 10,240 and 5,120 samples, i.e., 10% and 5% of the default training size, respectively[6]. The validation set is also reduced to 1,024 samples (10% default size), but the testing set size remains unchanged.

We observe two specifically interesting phenomenons: 1) for the first time in our experiments, LFISTA outperforms Dense-LISTA notably, and the margin seems to enlarge as the training size decreases; (2) the searched architecture largely outperforms the other three when the training size is 10,240; yet it becomes comparable to LFISTA at the smaller 5,120 training size, albeit still clearly surpassing LISTA and Dense-LISTA. The lessons we learned seem to convey compound information: (i) LFISTA's robust performance suggests that model-based unrolling indeed provides strong inductive prior, that is advantageous under data-limited training; (ii) compared to $[b - i]$ where Dense-LISTA has consistently strong performance, adding overly dense connections seems not to be favored in data-limited regimes; (3) our searched architecture appears to be the right blend of model-based prior and data-driven capacity, that also possesses robustness to data-limited training.

## 4 OPEN THE BOX: WHAT GOOD DESIGN PATTERNS ARE INSIDE?

**A closer look at the averaged connectivity.** Experiments in Section 3 indicate that significantly improved architectures can be spotted by NAS, whose strong performance generalizes in many unseen settings. We now dive into analyzing our searched average architecture. Figure 2 plots the average of the top-50 architectures *before threshold*, i.e., each value indicate the "percentage" of top-50 models having that connection (1 denotes all and 0 none). The following patterns are observed:

- The consensus on connectivity is overall high. There are nearly 50% connections that over 90% of the top-50 models agree not to use. It seems the sparse recovery task obeys certain model-based prior and does not favor overly densely connections, as seen in Dense-LISTA.

---

[5]Many applications (such as video-based) can be formulated as sparse coding models whose dictionaries are subject to small dynamic perturbations (e.g, slowly varied over time) (Liu et al., 2019; Zhao et al., 2011).

[6]We did not go smaller since all performance would catastrophically drop when training size is below 5%.

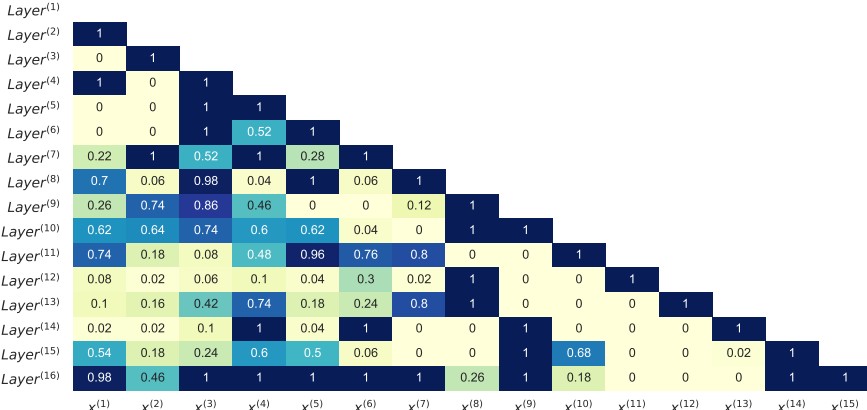

Figure 2: Visualization of our searched "averaged architecture". The vertical and horizontal axes denote the "destination" and "origin" layer indices for a skip connection. Note that every layer is always taken as input for its immediate preceding layer, therefore the diagonal line is all one.

- Besides the default connection (diagonal line), every layer uses at least one extra skip connection. If counting in thresholding (0.5), all layers except layers 12, 14, 15 "activate" no less than 1/3 possible input skips, and more than half layers "activate" no less than 1/2. Those extra connections might represent or be interpreted as acceleration terms, potentially more sophisticated than studied in (Moreau & Bruna, 2017): we leave for future research.

- Early and late layers have better consensus on connections, while middle layers (9-11) display some diversity of choice. The last layer (16) looks particularly special: it has connections from most preceding layers with high agreements. Our further experiments find that those connections to the last layer contribute a notable part to our searched performance: if we only pick those extra connections to layer-16 from our average architecture, and add them to LISTA/LFISTA, then we can immediately boost LISTA from -43.3 dB to -48.2 dB, and LFISTA from -47.3 dB to -48.4 dB (noiseless setting). Understanding the role of "densely connected last layer" in unrolling seems to be an interesting open question.

**Transferring found pattern to other unrolling.** We briefly explore whether the LISTA-found design patterns can generalize to other unrolling. We choose differentiable linearized ADMM (D-LADMM) (Xie et al., 2019), another competitive unrolling model for sparse recovery, as the test bed. We directly transplant the connectivity in Figure 2, by adding those extra (non-diagonal) connections to augment D-LADMM, with no other change. We compare (a) the original D-LADMM; (b) random architectures with 42 extra connections sampled i.i.d. (the same total number as the transferred pattern); (c) D-LADMM with dense connections added only to the latter half layers (77 extra connections in total); (d) a densely-connected D-LADMM constructed similar to Dense-LISTA with 105 extra connections; (e) our transferred one with 42 extra searched connections. For the random connected models, we sample 5 patterns and report the average NMSEs.

The three models are trained and evaluated in three representative settings: (1) noiseless: the three models' NMSEs are -54.2 / -55.0 / -55.3 / -55.4 / -55.6dB, respectively; (2) noisy (SNR = 20): NMSEs -18.2 / -18.9 / -18.9 / -19.0 / -19.1 dB; (3) data-limited (training size 10,240): NMSEs -24.8 / -25.6 / -29.8 / -22.6 / -33.5 dB. Several observations and conclusions can be drawn:

- The transferred connectivity pattern immediately boosts D-LADMM in all three settings;

- The more dense variants perform comparably with our augmented one when training data is sufficient, but degrades even behind the original in the data-limited regime;

- The *Dense-in-Latter-Half* does not outperform the transferred pattern, and especially degrades the performance when the training data becomes limited. It seems some earlier-layer extra connectivity might help the trainability of the early layer in those cases. That indicates that our learned connectivity pattern is more subtle than naively "denser for latter";

- The randomly connected models with the same amount of extra connections as the transferred one are in general on par with the *Dense-in-Latter-Half*, but even worse on limited data. Those show that although appropriate connection percentage is a useful factor, it does not constitute the major value of our searched specific connectivity.

## 5 DISCUSSIONS AND FUTURE WORK

While unrolling often yields reasonably good architectures, we seem to discover consistently better networks in this study. But *are we denying the advances and promise of the unrolling field?* **Absolutely Not**. As veterans studying unrolling, we hope this work to provide a reference and spark more reflections, on when unrolling is useful, how to improve it more, and (broadly) how to mingle model-based optimization and data-driven learning better. A good perspective was taken in (Monga et al., 2019), suggesting that with the original model-based optimization (Dittmer et al., 2019) induced as a prior, the unrolled architectures behave as "an intermediate state between generic networks and iterative algorithms", possessing relatively low bias and variance simultaneously. Unrolled networks might be more data-efficient to learn (supposed by our data-limited training experiments). Meanwhile if training data is sufficient, or if linear sparse model is not the accurate prior, then unrolling does not have to be superior. We then recommend data-driven model search or selection, perhaps leveraging the unrolled model as a robust starting point in the design space.

We shall further comment that this study is in a completely different track from theoretically understanding LISTA as a learned optimization algorithm (Moreau & Bruna, 2017; Giryes et al., 2018; Chen et al., 2018; Liu et al., 2019; Aberdam et al., 2020). Those works' interests are mainly on interpreting the unrolled architecture as an early-truncated iterative optimizer (with adaptive weights): unrolling plays the central role in bridging this optimization-wise interpretability. Unrolling can also connect to more desired stability or robustness results from the optimization field (Aberdam et al., 2020; Heaton et al., 2020), which makes great research questions yet is beyond this paper's scope.

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

## A APPENDIX

### A.1 HAND-CRAFTED CONNECTIVITY BASELINES

Other than the original unrolled LISTA, we also compare our searched connectivity pattern with two hand-crafted connectivity baselines LFISTA and Dense-LISTA. Figure 3 shows a straightforward visualization for these two hard-crafted connectivity patterns.

### A.1.1 LFISTA

The formulation of FISTA (Beck & Teboulle, 2009) is

$$
\begin{aligned}
t^{(k+1)} &= \frac{1 + \sqrt{1 + 4t^{(k+1)^2}}}{2} \\
y^{(k+1)} &= x^{(k)} + \frac{t^{(k)} - 1}{t^{(k+1)}}(x^{(k)} - x^{(k-1)}) \\
x^{(k+1)} &= \eta_{\lambda/L}\left(y^{(k+1)} + D^T(b - Dy^{(k+1)})/L\right).
\end{aligned}
\tag{7}
$$

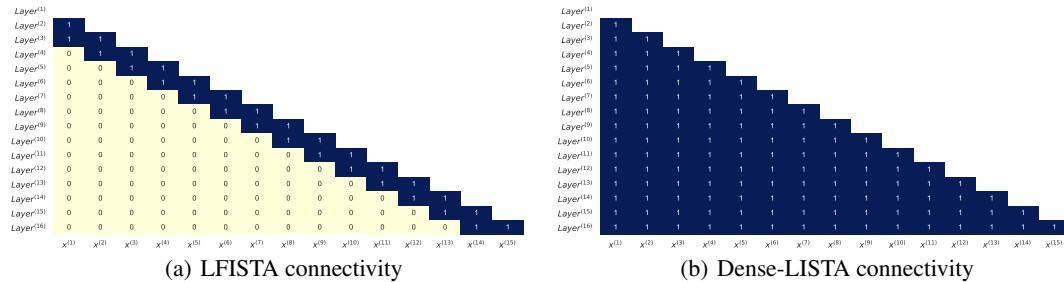

(a) LFISTA connectivity  (b) Dense-LISTA connectivity

Figure 3: Connectivity visualization for LFISTA and Dense-LISTA. The vertical and horizontal axesdenote the "destination" and "origin" layer indices for a skip connection. 1 means the connection is activated.

It is natural to parameterize the second step as

$$y^{(k+1)} = c_1^{(k)} x^{(k)} + c_2^{(k)} x^{(k-1)}, \tag{8}$$

which will reduce FISTA to LWA that is introduced in the main text. Of course we can plug $y^{(k+1)}$ into the third equation in (7), re-arrange and combine terms and parameterize the iteration as

$$x^{(k+1)} = \eta_{\lambda/L} \left( W_1^{(k+1)} x^{(k)} + W_2^{(k+1)} x^{(k-1)} + W_b b \right).$$

However, this will introduce more parameters than LWA. As we mainly consider LWA in the main text, we use the parameterization in (8) in this paper for fair comparison. Introducing FISTA to our design space would add 14 extra skip connections.

### A.1.2 DENSE-LISTA

Inspired by successful application of dense connections in deep learning (Huang et al., 2017), we manually enable all possible skip connections in our connectivity design space. For this densely-connected LISTA instantiation, each layer will take the outputs of all its previous layers as input, leading to 105 extra skip connections comparing to the original LISTA.

### A.2 COMPLEMENTARY EXPERIMENTS

### A.2.1 DIFFERENT PROBLEM DIMENSIONALITIES

We are also curious whether our finding is also applicable to sparse coding problems with different dimensionalities. Here we follow the same noiseless setting as Section3.2 in the main text, but enlarge the dictionary to $512 \times 1024$, i.e. $m = 512, n = 1024$, and regenerate the training, valiation and test data with the new dictionary. As shown in the first row of Table 2, our results with new dimensionality setting are highly consistent to the noiseless measurement setting, indicating our searched pattern does not overfit to a specific problem size.

### A.2.2 ILL-CONDITIONED DICTIONARIES

Most dictionaries used in the experiments are well-conditioned Gaussian random matrices that enjoy ideal properties such as natural incoherence. However, in real-world applications, this is too ideal to be true. Here we manually create ill-conditioned matrices and use them as the dictionaries. We sample two Gaussian matrices $U \in \mathbb{R}^{250 \times 200}$ and $V \in \mathbb{R}^{200 \times 500}$. Then we use $D = UV$ as the dictionary, which is inherently low-rank. With an ill-conditioned dictionary, our searched architecture still outperforms other counterparts as demonstrated in the second row of Table 2.

### A.2.3 REAL-WORLD COMPRESSIVE SENSING

Beyond synthesis experiments with generated signals with ideal sparsity, we further re-used the architecture searched on the synthetic data, and train/test this architecture on Natural Im-

Table 2: Complementary experiment results for different dictionaries.

| Dictionary | LISTA | LFISTA | Dense-LISTA | Searched |
|---|---|---|---|---|
| $D \in \mathbb{R}^{512 \times 1024}$ | -44.1 | -47.0 | -47.9 | **-52.4** |
| $D = U \in \mathbb{R}^{250 \times 200} \cdot V \in \mathbb{R}^{200 \times 500}$ | -27.1 | -33.4 | -42.4 | **-42.8** |

age Compressive Sensing, following the setting in Section 4.2 of (Chen et al., 2018). We extract $16 \times 16$ patches from the natural images in BSD500 dataset and downsample the patches to 128-dimension measurements using a $128 \times 256$ Gaussian sensing matrix (e.g., compressive sensing ratio = 50%). The dictionary ($256 \times 512$) is learned using a dictionary learning algorithm (Xu & Yin, 2013). We train LISTA, LFISTA, Dense-LISTA and the searched architecture on 400 training images on BSD500 and test the trained model on 11 standard testing images (Kulkarni et al., 2016). The results are shown in Table 3 where the average PSNR in decibel on the testing images are reported. We run three repetitions of training for all architectures and report the average PSNR over the three runs. We observe that although the architecture is searched on a synthetic data/task and then directly reused for training on a different new task on new data (natural image patches), the searched architecture still performs robustly, outperforming LISTA by 0.11 dB, and slightly surpassed LFISTA and Dense-LISTA.

Note that the synthetic data and natural images have significant gaps. Although the transferred architecture is not dedicatedly searched for the compressive sensing problem, it still performs very robustly; a re-searched architecture on natural images would naturally only expect better performance.

Table 3: Real-world compressive sensing experiment results in PSNR (dB) on our searched average architecture (from Section 3.2) and three baselines.

| Model | LISTA | LFISTA | Dense-LISTA | Searched |
|---|---|---|---|---|
| PSRN (dB) | 34.67 | 34.73 | 34.69 | 34.78 |

## A.3 CONNECTION INJECTION OPTIONS

Other than the *Learnable Weighted Average* (LWA) we use by default in the main text, we also consider two other options to inject new connections.

### A.3.1 NAIVE AVERAGE (NA)

The most naive approach to fuse extra connectivity into the unrolled LISTA (4) is to simply take the average of the outputs of all chosen layers, before we apply the linear transform $\mathbf{W}^{(k)}$. Formally, we replace $\mathbf{x}^{(k)}$ in (4) with

$$\tilde{\mathbf{x}}^{(k)} = \frac{1}{\sum_{i=1}^{k} g_{i,k}} \sum_{i=1}^{k} g_{i,k} \cdot \mathbf{x}^{(i)}, \tag{NA}$$

where $g_{i,k} \in \{0, 1\}$ is a gate that controls whether the $i$-th iterate is chosen as the input to the $k$-th layer (same hereinafter). Denote the gates that connect the $k$-th layer, e.g. all gates in (NA) as vector $\mathbf{g}_k = (g_{1,k}, \dots, g_{k,k})^T$, and then concatenate into a long ordered vector $\mathbf{g} = \left[\mathbf{g}_1^T, \dots, \mathbf{g}_{K-1}^T\right]^T$. All possible combination of the gate values of the gates form the search space $\mathcal{H} = \{\mathbf{g}\}$.

### A.3.2 MOMENTUM (MM)

We also consider a more complicated way to manipulate extra skip connections, inspired by momentum approaches Sutskever et al. (2013) in optimization. Several acceleration approaches for ISTA leverage momentum terms, such as the renowned Fast ISTA (FISTA) Beck & Teboulle (2009); Bioucas-Dias & Figueiredo (2007). To allow for flexibly learnable momentum, we illustrate our idea with a simple example, where only last two iterates are considered in the momentum. In this way, the ISTA iteration (3) is extended as:

$$\mathbf{x}^{(k+1)} = \eta_{\theta^{(k)}}(\mathbf{x}^{(k)} + \beta^{(k)}\mathbf{v}^{(k)}). \tag{9}$$

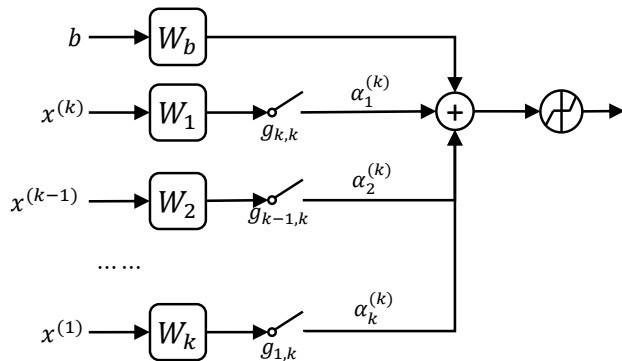

Figure 4: An illustration of connection injection with Momentum.

where $\beta^{(k)}$ is a hyperparameter that controls the strength of momentum. Note that (9) reduces to ISTA (3) if we set $\beta^{(k)} \equiv 1/L$, $\theta^{(k)} \equiv \lambda/L$ and $\mathbf{v}^{(k)} = \nabla_{\mathbf{x}}f(\mathbf{x}^{(k)}) = \boldsymbol{D}^T(\mathbf{b} - \mathbf{D}\mathbf{x}^{(k)})$. $\mathbf{v}^{(k)}$ is the update direction that already takes momentum into consideration involving the last two iterations:

$$\mathbf{v}^{(k)} = \gamma^{(k)}\nabla_{\mathbf{x}}f(\mathbf{x}^{(k)}) + (1 - \gamma^{(k)})\nabla_{\mathbf{x}}f(\mathbf{x}^{(k-1)}) \tag{10}$$

$$= \gamma^{(k)}\mathbf{D}^T(\mathbf{b} - \mathbf{D}\mathbf{x}^{(k)}) + (1 - \gamma^{(k)})\mathbf{D}(\mathbf{b} - \mathbf{D}\mathbf{x}^{(k-1)} \tag{11}$$

$$= \mathbf{D}^T\mathbf{b} - \gamma^{(k)}\mathbf{D}^T\mathbf{D}\mathbf{x}^{(k)} - (1 - \gamma^{(k)})\mathbf{D}^T\mathbf{D}\mathbf{x}^{(k-1)} \tag{12}$$

Substituting (12) in (9) gives

$$\mathbf{x}^{(k+1)} = \eta_{\theta^{(k)}}\left(\beta^{(k)}\mathbf{D}^T\mathbf{b} + (\mathbf{I} - \beta^{(k)}\gamma^{(k)}\mathbf{D}^T\mathbf{D})\mathbf{x}^{(k)} + (-\beta^{(k)}(1 - \gamma^{(k)})\mathbf{D}^T\mathbf{D})\mathbf{x}^{(k-1)}\right). \tag{13}$$

Denote $\hat{\mathbf{W}}_{\mathbf{b}}^{(k)} = \beta^{(k)}\mathbf{D}^T$, $\hat{\mathbf{W}}_{1}^{(k)} = \mathbf{I} - \beta^{(k)}\gamma^{(k)}\mathbf{D}^T\mathbf{D}$ and $\hat{\mathbf{W}}_{2}^{(k)} = -\beta^{(k)}(1 - \gamma^{(k)})\mathbf{D}^T\mathbf{D}$, and then we get the untied version (do not share $\hat{\mathbf{W}}_{\mathbf{b}}^{(k)}, \hat{\mathbf{W}}_{1}^{(k)}, \hat{\mathbf{W}}_{2}^{(k)}$ within layers) of (8):

$$\mathbf{x}^{(k+1)} = \eta_{\theta^{(k)}}(\hat{\mathbf{W}}_{\mathbf{b}}^{(k)}\mathbf{b} + \hat{\mathbf{W}}_{1}^{(k)}\mathbf{x}^{(k)} + \hat{\mathbf{W}}_{2}^{(k)}\mathbf{x}^{(k-1)}), \tag{14}$$

As suggested by the seminal work (Liu et al., 2019), sharing the above weight matrices (corresponding to using layer-invariant $\beta$ and $\gamma$) across layers does no hurt to the performance while reducing the parameter complexity. However, we introduce two scaling parameters $\alpha_1^{(k)}$ and $\alpha_2^{(k)}$ (will be trained using data) to loosen the constraint, yielding a simpler form of (14):

$$\mathbf{x}^{(k+1)} = \eta_{\theta^{(k)}}(\hat{\mathbf{W}}_{\mathbf{b}}\mathbf{b} + \alpha_1^{(k)}\hat{\mathbf{W}}_1\mathbf{x}^{(k)} + \alpha_2^{(k)}\hat{\mathbf{W}}_2\mathbf{x}^{(k-1)}). \tag{15}$$

Extending (15) to the general case where all previous iterates could be chosen by a gate to get involved in the momentum term $\mathbf{v}^{(k)}$, we get

$$\mathbf{x}^{(k+1)} = \eta_{\theta^{(k)}}\left(\hat{\mathbf{W}}_{\mathbf{b}}\mathbf{b} + \sum_{i=1}^{k} g_{k+1-i,k} \cdot \left(\alpha_i^{(k)}\hat{\mathbf{W}}_i\mathbf{x}^{(k+1-i)}\right)\right), \tag{MM}$$

where the subscript in $\hat{\mathbf{W}}_i$ means its input to come from the $i$-th last iterate and $\alpha_i^{(k)}$ is the step size parameter associated with $\hat{\mathbf{W}}_i$ in the $k$-th layer. Step sizes $\alpha_i^{(k)}$ are initialized with 1.

### A.3.3 EMPIRICAL RESULTS

Before launching our large-scale search experiment, we first compare three of our connection injection methods (LWA, NA and MM) by randomly sampling about 500 architectures from our design space, and train each of them individually following our single model training protocol in Section 3. Since we are focusing on evaluating injection methods, we use soft-thresholding as our neuron type by default.

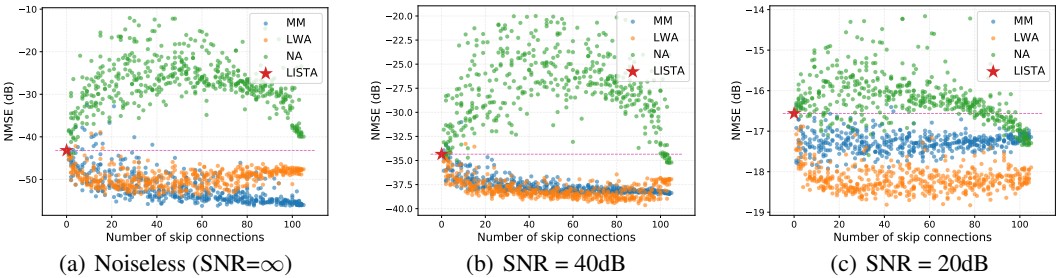

|   |   |   |
|---|---|---|
| (a) Noiseless (SNR=∞) | (b) SNR = 40dB | (c) SNR = 20dB |

Figure 5: The NMSE distribution of sampled architectures, using NA, LWA and MM to inject extra skip connections, in synthetic experiments under different SNRs. The red star marks the original LISTA. x-axis from left to right denotes the number of added extra connections from low to high. y-axis denotes the models' obtained NMSEs (the lower the better). Best zoom in and view in color.

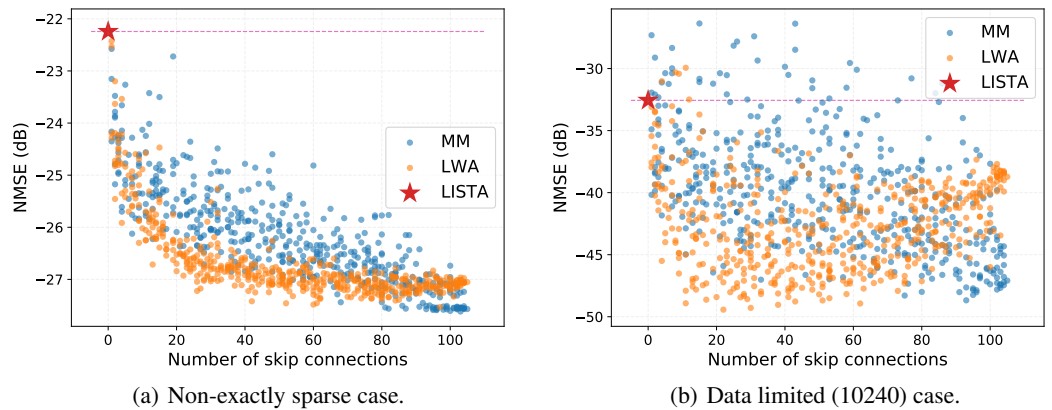

|   |   |
|---|---|
| (a) Non-exactly sparse case. | (b) Data limited (10240) case. |

Figure 6: Distribution of sampled architectures in non-exactly sparse vector case and limited training data case.

**Noiseless measurement and Gaussian noise.** We first conduct experiments under three most common settings (noiseless measurement, noisy measurement with 40dB and 20dB additive Gaussian noise). As shown in Figure 5, NA always hurts the performance in all three settings. While MM could perform slightly better than LWA in noiseless setting, LWA clearly beats MM in noisy setting for both SNR=40dB and SNR=20dB. Based on our observation that NA constantly yields worse performance, we conduct the rest comparative experiments using only LWA and MM.

**More complicated settings.** Other than additive Gaussian noise cases, we also evaluate and compare LWA and MM on the two more challenging and realistic settings: non-exactly sparse vector and limited training data. We follow the same setting as described in Section 3.3. We use 10240 as training data size in limited training data setting. From NMSE distributions in both Figure 6 and noisy setting in Figure 5, we can clearly tell that LWA has better robustness when the same connectivity pattern is applied. At the same time, probably due to the larger number of parameters introduced, MM suffers more from limited training data.

Since MM only wins in the easiest noiseless case with sufficient training data, we can safely tell that LWA has more practical values as it is consistently the winner of settings where we have different noise in the data and where the data is limited. Since we are not in an ideal world, we choose to conduct all our main experiments using LWA to fuse injected connections.

### A.4 SIDE CONNECTION PRUNING

Despite of our focus on adding skip connections in this paper, we also empirically evaluate what will happen if we also allow for pruning connections. As shown in Figure 7(a), we insert gate functions to side connections to decide whether they will be removed from the original unrolled LISTA. Specifically, we allow all the side connections except the first one to have the probability being removed from the architecture. As we use $K = 16$ LISTA layers as our default setting, our pruning search space is consisted of $2^{15}$ potential candidates for original unrolled LISTA architecture. For our joint injection and pruning setting, the search space is enlarged to $2^{105} \times 2^{15} = 2^{120}$. Note we also utilize the tool of random sampling here as in Section A.3.3 and choose to use soft-thresholding for all layers.

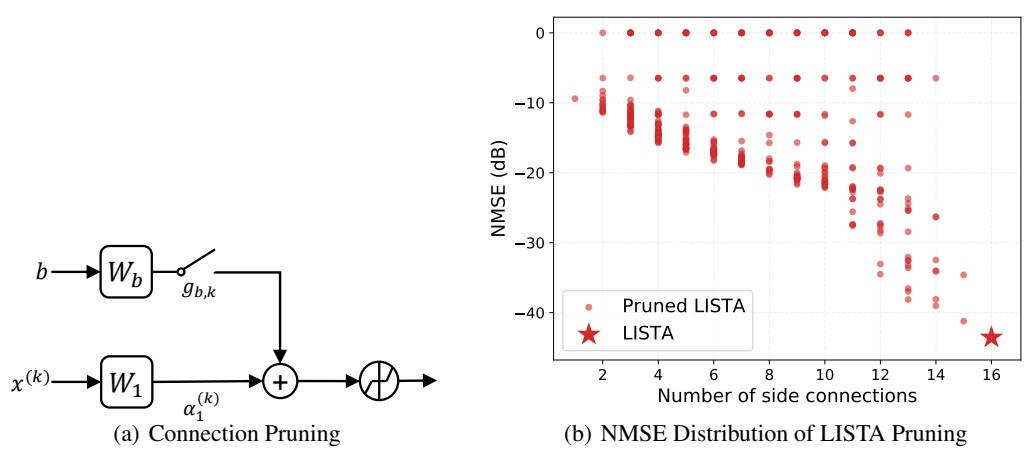

(a) Connection Pruning     (b) NMSE Distribution of LISTA Pruning

Figure 7: Pruning from original unrolled LISTA.

#### A.4.1 PRUNING FROM ORIGINAL LISTA

We first apply the side connection pruning search space to the original LISTA. As shown in the Figure 7(b), removing side connections will **always** do harm to the unrolled LISTA model, which echos the finding observed by (Wang et al., 2016a).

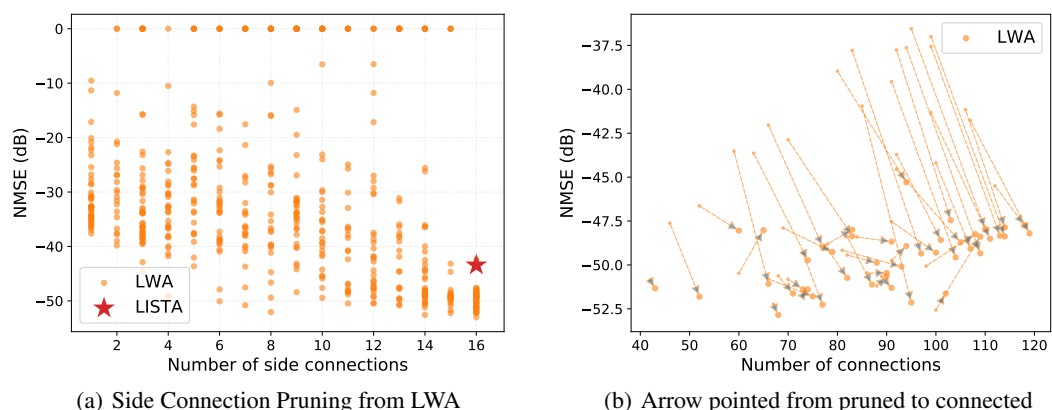

(a) Side Connection Pruning from LWA   (b) Arrow pointed from pruned to connected

Figure 8: Pruning from LWA architectures.

#### A.4.2 JOINT INJECTION AND PRUNING

Next we apply both connection injection and side connection removal at the same time. Our results in Figure 8(a) illustrate that although connection injection can greatly reduce the influence of side

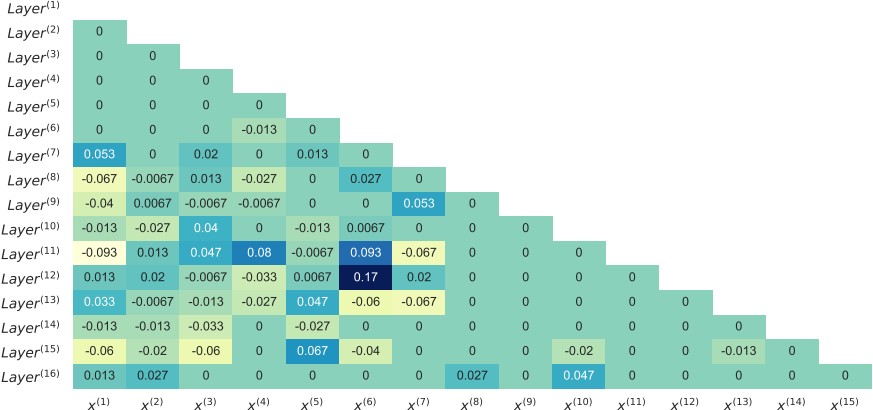

Figure 9: The connection difference between the default top-50 average and top-30 average.

connection pruning, most architectures from this design space suffer from the removal of side connections. We also randomly select some models in this setting, keep their injected skip connections and reconnect all their side connections. Figure 8(b) shows how NMSEs change for LWA models. The arrows point from the "pruned side connections" case to "full side connections" case and the x axis means the total number of skip connections and side connections. As we can see, we can almost always gain performance by reconnecting all side connections, which is aligned with our observations in A.4.1. Our observation here indicates that we should always keep all side connections.

## A.5 STABILITY OF AVERAGED ARCHITECTURE

To further demonstrate the stability of our averaged architecture, we recalculate an averaged architecture from top-30 models, and compute its connection differences with the default top-50 average one. The difference map is visualized in the Figure 9. Clearly, they are highly aligned and echo our observations that those top architectures already have high agreement on connections to be activated.

