# OpenReview forum: "A Design Space Study for LISTA and Beyond"
_ICLR.cc/2021/Conference — ICLR 2021 Poster_

### Official Review · AnonReviewer4 · 2020-10-27
**An interesting idea but the conclusions of this work are not very significant**

**Rating:** 4
**Confidence:** 3

**Review:**

Summary of contributions:

In this paper, authors performed a neural architecture search to improve LISTA algorithm for solving the lasso on synthetic data. The main motivation of this paper is to investigate the effectiveness of unrolling in comparaison with a more "black box" architectures. The "averaged" top architectures found by the study give better performance than LISTA when the  two models are trained on the same dataset.

Strengths:

-The idea of using NAS to study unrolled models seems to be novel to the best of my knowledge.

-This idea is interesting. I definitely agree that the relevance of unrolled architectures should be questioned.

-The authors did a good job to describe clearly their idea and the experiments they conducted (at the exception of the last paragraph "transferring found patterns to other unrolling" which lacks clarity in my opinion).

Weaknesses:

-I think that the overall conclusions of the paper are not very significant. The authors only focus on a very simple problem : the lasso on synthetic data (with known dictionaries). In such a simple setting I do not see why one would consider using this trainable model rather than more standard algorithms for solving the lasso (which are significantly faster than ISTA and have convergence guarantees and/or duality gap).

-I am not convinced by the results regarding transferability. Even if the searched model outperforms the other methods in the simplest setting, the searched model seems to be very sensitive to perturbations. Looking at line (i) of table 1 with perturbed dictionary, the gap between searched model and LFISTA is small and the searched model performs as good as Dense Lista. More generally, the searched model seems to perform as good as Dense LISTA in most of the perturbed settings. Again this question the utility of this model.

-The authors only consider the problem where the underlying optimization problem is known. I think it would be much more interesting to study the case when this underlying optimization problem is not known and has to be learnt (that aspect is briefly discussed at the end of the related work section). For example what if the dictionary is not known ? Lista offers the possibility to learn the dictionaries with end-to-end training. What would be the results of the NAS in the setting with unknown dictionary, how would unrolled models perform in that setting ?

Suggestions to authors:

-It would be interesting to consider the case where the underlying optimization problem is not known a priori. In that situation I think that unrolled models make more sense  (unrolled weighted-l1sparse coding for a simple task such as denoising, experiments with learned dictionaries ...). In that setting I would be more curious regarding the results of the NAS study.

-I think it would be relevant to compare the unrolled models to other algorithms for solving the lasso (lars...). In my opinion, it would clarify the conclusion of the paper. (A comparison in term of inference speed, accuracy, computational cost would be interesting).

---

> ### Author Response · Authors · 2020-11-20
> **Response to AnonReviewer4**
>
> We thank the reviewer for the constructive comments and feedback. Below, we provide detailed responses to your concerns.
>
> Q1: The overall conclusions of the paper are not very significant.
>
> A1: We humbly yet firmly disagree on “In such a simple setting I do not see why one would consider using this trainable model rather than more standard algorithms for solving the lasso”. Reasons are two-folds:
>
> 1. Solving LASSO or sparse recovery with known dictionaries has been an important playground for many previous unrolling works in the literature, and keeps drawing attention from the optimization and digital signal processing community. Numerous papers have taken synthetic sparse recovery (with known dictionary) as a common testbed, to assess the effectiveness of their new unrolled models, e.g. (Moreau & Bruna, 2017; Giryes et al., 2018; Chen et al., 2018; Liu et al., 2019; Ablin et al., 2019). Practical applications of such settings are widely found in signal processing, e.g. please check https://arxiv.org/pdf/1906.05774.pdf just for example. Therefore, we are following a common, standard, widely-adopted and practically meaningful testbed.
> 2. On the problem and data distribution that has been seen during training (i.e. there is no training-testing mismatch), LISTA commonly takes 10-20 steps to find a good solution, while ISTA takes thousands of steps. We are familiar with various sparse solvers, some of which are indeed “faster than ISTA and have convergence guarantees and/or duality gap“. However, we are aware of NO such standard solver who can match the 10-20 step cost of LISTA (on seen distribution). For example, (Chen et al., 2018) provided fair comparison between LISTA with AMP and FISTA, proving the former to be the clear winner (Figure 5). We thus think that should leave no confusion why a LISTA solver is preferred over those standard ones, when one needs to solve such similar problems repeatedly - and this should be a commonly-accepted motivation and ground for the whole unrolling research field.
>
> Q2: Transferability
>
> A2: We humbly suggest that the reviewer might miss the point here. While it has been explained that our setting is a standard and popular one, we have even significantly augmented our evaluation to be more comprehensive and rigorous, compared to all existing works mentioned above. We lay out out abundant comparisons in the (not usually inspected) cases of cross-noise (level and type), perturbed dictionary, non-exact sparsity, and limited training data. We suggest that our new evaluation protocols may be of independent interest to the newly designed models in the unrolling field, and are another evidence our setting is really not “simple”.
>
> Our transferability is then stated on the key fact that,  along all those new diverse evaluation dimensions, the searched architecture is consistently superior, even just searched once under one setting. Note that winning-all is highly non-trivial. For example in Table 1. LFISTA and Dense-LISTA are both architectures that are derived from certain priors. LFISTA is derived by unrolling FISTA algorithm and Dense-LISTA comes from the inductive bias in vision tasks that densely connected convolutional networks usually perform better. For cases where the training data is sufficient (from exp [b] to exp [i]), Dense-LISTA is consistently better than LFISTA and our conjecture is that Dense-LISTA has more trainable parameters and thus larger capacity than LFISTA. However, in the data-limited cases, i.e. exp [j] and [k], LFISTA outperforms Dense-LISTA by obvious gaps.
>
> Then, looking at our searched architecture, it performs always better (except only two cases on par) than both LFISTA and Dense-LISTA by large gaps in those settings. Overall, we believe that just one setting on tie with the Dense LISTA baseline really does not “question the utility of this model”: it shouldn’t be overlooked that our searched architecture demonstrate such impressive full-stack utility/transferability in all those settings, which is a proof of our effective search.

---

> ### Author Response · Authors · 2020-11-20
> **Response to AnonReviewer4 (Continued)**
>
> Q3: The authors only consider the problem where the underlying optimization problem is known.
>
> A3: We first, once again, emphasize that our current setting is standard, effective, non-trivial, and reveals the critical insights that we target for. To clarify a possible confusion here, we do NOT see our current setting has explicitly taken advantage of “known dictionary” besides end-to-end training. Please note that we train every sampled model from scratch, without parameterizing the model weights using the dictionary, nor even using the known dictionary as initialization. In other words, our sampled model training is totally blind to the dictionary too, except seeing training pairs generated with the default dictionary.
>
> Second, we could consider re-doing our whole search on natural images, but it is impractical to complete in a short rebuttal time window. To answer your curiosity, we directly re-used the searched architecture on the synthetic data, and train/test this architecture on Natural Image Compressive Sensing, following the setting in Section 4.2 of [1]. We extract 16x16 patches from the natural images in BSD500 dataset and downsample the patches to 128-dim measurements using a 128x256 Gaussian sensing matrix (e.g., CS ration = 50%). The dictionary (256x512) is learned using a dictionary learning algorithm [2]. We train LISTA, LFISTA, Dense-LISTA and the searched architecture on 400 training images on BSD500 and test the trained model on 11 standard testing images [3]. The results are shown below where the average PSNR on the testing images are reported. We run three repetitions of training for all architectures and report the average PSNR over the three runs. We observe that although the architecture is searched on a synthetic data/task and directly trained for a different new task (compress sensing) on new data (natural image patches), the searched architecture still performs robustly, outperforming LISTA by 0.11 dB, and slightly surpassed LFISTA and Dense-LISTA. Note that the synthetic data and natural images have significant gaps. Although the transferred architecture is not dedicatedly searched for the compressive sensing problem, it still performs very robustly; an re-searched architecture on natural images would naturally only expect better performance.
>
> We hope our above two points have addressed all your possible concerns on this point.
>
>
> | Architecture             | Compressive Sensing PSNR |
> |--------------|:--------------------------:|
> | LISTA        | 34.67               |
> | LFISTA       | 34.73               |
> | Dense-LISTA  | 34.69           |
> | Searched     | 34.78             |
>
>
> Q4: Compare the unrolled models to other algorithms for solving the lasso (lars...)
>
> A4: We apply LARS to the testing data in the noiseless setting, i.e. the testing data used in experiment [b] in Table 1. We observe that LARS achieves an average of -17.47dB NMSE at the 50th iteration; and takes ~ 100 steps on average to achieve -40dB NMSE. In contrast, the vanilla LISTA achieves -43.3 dB and the searched architecture -54.2dB both with 16 layers. During inference, LISTA and its variants have similar computation cost and inference speed as 16 iterations of ISTA algorithm. The author team is experienced in various sparse optimization algorithms, and we find those results no surprise to us, as already discussed in A1.
>
>
> [1] Xiaohan Chen, Jialin Liu, Zhangyang Wang, and Wotao Yin. Theoretical linear convergence of unfolded ista and its practical weights and thresholds. In Advances in Neural Information Processing Systems, pp. 9061–9071, 2018.
>
> [2] Yangyang Xu and Wotao Yin. A block coordinate descent method for regularized multiconvex optimization with applications to nonnegative tensor factorization and completion. SIAM Journal on imaging sciences, 6(3):1758–1789, 2013.
>
> [3] Kuldeep Kulkarni, Suhas Lohit, Pavan Turaga, Ronan Kerviche, and Amit Ashok. ReconNet: Non-iterative reconstruction of images from compressively sensed measurements. In Proceedings of the IEEE Conference on Computer Vision and Pattern Recognition, 2016

---

### Official Review · AnonReviewer3 · 2020-10-27
**A good submission**

**Rating:** 7
**Confidence:** 4

**Review:**

This paper studied the structure designing problem for a class of neural networks, i.e., the unrolling networks, or more specifically the LISTA model. Specifically, the authors varied the connectivity patterns and neuron types to define a rich searching space for the network design, and then applied the neural architecture search techniques to fined the best networks.

The studied problem is interesting and important, and the conducted experiments reveals some insights of unrolling based deep networks, which provides some useful guidelines for further research in this direction. The paper is also well written and easy to follow. Overall, I think this is a good submission.

A drawback, concerning about the experiments, of the manuscript is that it lacks applications. Therefore, a suggestion is to apply the learned network structures to more practical problems, such as image denoising and compressive sensing, or something else, to see whether they are effective in practice.

---

> ### Author Response · Authors · 2020-11-20
> **Response to AnonReviewer3**
>
> We thank the reviewer for the positive review and the constructive feedback. We follow your suggestion to conduct the experiment on Natural Image Compressive Sensing, following the setting in Section 4.2 of [1]. We extract 16x16 patches from the natural images in BSD500 dataset and downsample the patches to 128-dim measurements using a 128x256 Gaussian sensing matrix (e.g., CS ratio = 50%). The dictionary (256x512) is learned using a dictionary learning algorithm [2].
>
> We train LISTA, LFISTA, Dense-LISTA and the searched architecture on 400 training images on BSD500 and test the trained model on 11 standard testing images [3]. The results are shown below where the average PSNR on the testing images are reported. We run three repetitions of training for all architectures and report the average PSNR over the three runs. We observe that although the architecture is searched on a synthetic data/task and directly trained for a different new task (compress sensing) on new data (natural image patches), the searched architecture still performs robustly, outperforming LISTA by 0.11 dB, and slightly surpassed LFISTA and Dense-LISTA. Our future work will consider re-doing our search on natural images.
>
>
> | Architecture             | Compressive Sensing PSNR |
> |--------------|:--------------------------:|
> | LISTA        | 34.67               |
> | LFISTA       | 34.73               |
> | Dense-LISTA  | 34.69           |
> | Searched     | 34.78             |
>
> [1] Xiaohan Chen, Jialin Liu, Zhangyang Wang, and Wotao Yin. Theoretical linear convergence of unfolded ista and its practical weights and thresholds. In Advances in Neural Information Processing Systems, pp. 9061–9071, 2018.
>
> [2] Yangyang Xu and Wotao Yin. A block coordinate descent method for regularized multiconvex optimization with applications to nonnegative tensor factorization and completion. SIAM Journal on imaging sciences, 6(3):1758–1789, 2013.
>
> [3] Kuldeep Kulkarni, Suhas Lohit, Pavan Turaga, Ronan Kerviche, and Amit Ashok. ReconNet: Non-iterative reconstruction of images from compressively sensed measurements. In Proceedings of the IEEE Conference on Computer Vision and Pattern Recognition, 2016

---

### Official Review · AnonReviewer1 · 2020-10-29
**Contains extensive experiments but technical contributions are limited**

**Rating:** 6
**Confidence:** 4

**Review:**

This paper conducts an empirical study on unrolling architecture design, by applying NAS to search the connectivity pattern based on LISTA, and compares the searched model to the original LISTA.

Pros:
+ The paper is clearly written and not hard to follow
+ Experiments show that the searched architecture performs better than LISTA
+ The experiments are comprehensive, including various experimental settings that can compare the unrolled architectures from different aspects

Cons:
- Technically, it is a direct application of NAS to LISTA, so the methodological contribution is not very strong

Through extensive experiments, this paper shows the effectiveness of NAS for searching for a better architecture based on unrolled algorithms, via a case study on LISTA. It tells that unrolling may not be the best choice and better architecture can be found by NAS. However, the technical contribution and novelty of the approaches are not strong enough. I personally think this is a broader line paper and my score is actually between weak acceptance and weak rejection.

grammar:
- In the abstract, '.... and are able to discover networks that consistently better....' ->  '.... and are able to discover networks that ARE consistently better....'

====after author response===
I would like to thank the authors for the detailed response which resolves some of my concerns about the novelty of this paper. I agree that this is the first work (as far as I know) that brings NAS to unrolled algorithms. Unlike what I initially commented, 'my score is actually between weak acceptance and weak rejection', now I am happy to rate this paper as a weak acceptance.

---

> ### Author Response · Authors · 2020-11-20
> **Response to AnonReviewer1**
>
> We sincerely thank the reviewer for appreciating our comprehensive experiments, clear writeup, as well as the effectiveness of NAS for exploring the unrolling method. We also thank the reviewer for pointing out the grammar mistake in the abstract. We have fixed it in the updated version.
>
> Meanwhile, we respectfully disagree that our paper is just a “direct application of NAS to unrolling”.  We consider our study to be original and novel by itself, both methodologically and experimentally. Our reasoning is detailed below from three aspects, and we hope the reviewer could find it to more positively change his/her perception of our paper.
>
> ### We posed a novel question, and tried to answer it for the first time
>
> As AnonReviewer2 nicely summarized: “LISTA-style unrolling has been popular for deep learning-based inverse problems. But it is quantitatively unclear how good the unrolled models are, among all possible model variations. “ This is exactly the novel question that we posed, and tried to address for the first time.
>
> Although prior work vaguely suggests “unrolling may not be the best choice”, we are the first to quantitatively assess “how good the unrolled architectures actually are”, among all the possible design variations. Quoting also AnonReviewer3: “The studied problem is interesting and important, and the conducted experiments reveal insights of unrolling based deep networks, which provides some useful guidelines for further research in this direction.” We think an important merit of this work is to inspire broader reflections on unrolling as a design approach for deep networks.
>
> ### To use NAS for this question is novel. So is how to apply NAS.
>
> As we emphasize throughout this paper, “Our intention is not to innovate on NAS, but instead, to answer a novel question in LISTA using NAS as an experiment tool”. However, we respectfully disagree that our methodological contribution is discounted anyhow.
>
> First, to adopt NAS, one has to first define the design space of variations that we want to explore, starting from the unrolled model - which was never considered in this field before. The key questions include “what to vary”, and “how to vary”. For “what to vary”, we choose the connectivity patterns as well as neuron types, which are deemed as the most critical facts for unrolled architectures. For “how to vary”, the main question is how to introduce new connections. We introduced the most effective LWA way in Section 2.1, meanwhile also discussed what if we use naive averaging, momentum-based averaging, or even removing connections - in Appendix A.3 and A.4. Those resulted from an even larger-scale preliminary study and LWA was selected after careful comparison. As far as we know, no prior work ever considered how we could vary LISTA architectures, neither “what” nor “how”. Our work provides the first suites of solutions.
>
> Second, while our gigantic design space makes NAS-based exploration necessary, using NAS in this space has to be very careful to ensure result trustworthiness. We again quote Reviewer 2’s kind words: “Since NAS itself is not always stable, the authors present a number of efforts and discussions to support the reliability and reproducibility of their observations. Sec 2.3 shows their diligence in trying out multiple NAS algorithms; avoiding using weight sharing; and ‘averaging’ the top architectures to smooth out random fluctuations. I appreciate those valuable careful efforts.” Especially, the average architecture is an original novelty created by us (not seen in NAS before),  and helps us more robustly clearly perceive the “good patterns” shared by top models.
>
> We hope the reviewer could understand that such a large-scale study has no prior example nor established standard in the model unrolling field.  We have to build both design spaces (“what” and “how” to vary) and search strategies (how to ensure stability and trustworthiness), both from scratch. Our final methods presented are very thoughtfully selected, and hence convey notable intellectual merits by themselves. We appreciate AnonReviewer2 for pointing out: “bringing NAS to learning based algorithm design is a very novel and interesting direction. This paper can potentially turn into a seminal work here to inspire followers.”

---

> ### Author Response · Authors · 2020-11-20
> **Response to AnonReviewer1 (Continued)**
>
> ### Our experiments are based on careful design, that are both thoughtful and novel.
>
> Our experiments (Sections 3 and 4) went through the same rigorous design process. We ask more questions beyond what the conventional unrolled papers would typically ask, aiming for comprehensive evaluation about the pros and cons of the unrolled model variants. We would like to concretely understand what we can gain when we vary the connections, and meanwhile whether we pay any hidden price for the gain. Those contribute to our many new insights.
>
> For example, Table 1 lays out abundant comparisons in the (not usually inspected) cases of cross-noise (level and type), perturbed dictionary, non-exact sparsity, and limited training data. We indeed observe that there exist trade-offs in hand-crafted baselines in those cases, e.g., one can compare Dense LISTA versus LFISTA. Then, we can summarize that the searched architecture is indeed superior,  along all those new dimensions, confirming the all-round benefits of our searched architecture. We suggest that our new evaluation protocols may be of independent interest to the newly designed models in the unrolling field.
>
> We further present more visualization and analysis to “open the black box”, and find that the searched top architectures demonstrate highly consistent and potentially transferable patterns. Both the visualization of “averaged top architecture”, and the study of transferring the learned connectivity from one model family to another, are novel to not only the unrolling field, but also the NAS field.
>
> ### Summary
>
> In summary, we believe this study is a non-trivial one, both methodologically and experimentally. To address this novel interesting question for the first time, this study has to make *full-stack innovations* in defining the problem, formulating the space, solidifying the search strategy, rigorously designing experiments, and creatively analyzing results. We hope the reviewer could take them into account and more positively evaluate our work, for which we will be very grateful.

---

### Official Review · AnonReviewer2 · 2020-10-29
**NAS for unrolled LISTA**

**Rating:** 8
**Confidence:** 4

**Review:**

This paper studies a very interesting new problem of assessing unrolled models in a broader context using NAS methods. LISTA-style unrolling has been popular for deep learning-based inverse problems. But it is quantitatively unclear how good the unrolled models are, among all possible model variations. To fill in this gap, the authors first define a proper search space based on the varying connections and neurons from the unrolled LISTA backbone architecture. NAS is then exploited as the tool to find the best subsect of architecture from the large space.

This method is very intuitive yet solidly done. Since NAS itself is not always stable, the authors present a number of efforts and discussions to support the reliability and reproducibility of their observations. Sec 2.3 shows their diligence in trying out multiple NAS algorithms; avoiding using weight sharing; and “averaging” the top architectures to smooth out random fluctuations. I appreciate those valuable careful efforts.

The experimental study in Section 3 is very comprehensive and clearly laid out. Table 1 gives a great gist of their main findings. The authors compare with two natural baselines, LFISTA and Dense-LISTA. They prove NAS can indeed find superior architectures than those hand-crafted in all the different settings. The overall take-home point seems like no big surprise at first glance: unrolling provides a reasonably good architecture; yet if there are abundant training data, more complicated architectures can be discovered by varying connections (varying neurons seems not helpful).

But, a closer look at those experiments reveals many new finer-grained observations that really intrigue me. For example, it is interesting to see “the denser the better” is not the simple right claim in unrolling; and especially unrolling-based models are advantageous under data-limited training. Another good surprise is to see that all top architectures unanimously adopt soft thresholding as the only neuron type for all layers. The most interesting part of this paper is the "open the box" section 4 where the authors discuss the common characteristics of top architectures. Figure 2 is a highly interesting visualization and shows the top-50 architectures are indeed consistent on most connections.  It is very promising if we can understand further (1) why “the later the denser” - could that imply/be interpreted as switching to another higher-order algorithm in later iterations? and (2) why the last layer is particularly all-connected?

I think bringing NAS to learning based algorithm design is a very novel and interesting direction. This paper can potentially turn into a seminal work here to inspire followers. The paper is also well written, and the logic is very clear to follow. Section 5 concludes with a thoughtful discussion on what this work may connect and point to for the entire unrolling field.

I have a few suggestions and critiques for the authors to address:

-	Can we have a “weighted” version of Figure 2, say the higher-ranked architectures from the top-50 will get more weights when summed up; and in this way would we be able to observe more consensus of what relatively better models (if weighed more) tend to prefer?

-	LFISTA is a principled way to add more connections to LISTA using momentum. In this paper, the authors claimed their LWA to be better than momentum-based averaging in NAS. I wonder what will happen if replacing the momentum connection in LFISTA with the LWA way: will LFISTA performance improve or degrade?

-	I remain to be skeptical about the last LADMM example. It is rather unclear to me WHY the learned connectivity patterns from LISTA should transfer to LADMM? Perhaps, that implies some general idea of how dense connectivity could be properly injected in a sparse regression task. I would request to see comparison results from (1) adding dense connections only in the latter half layers; and (2) adding random connections, but with the same total connected percentage as the LISTA pattern. If either 1 or 2 can perform on par or better than the transferred LISTA pattern, the authors should consider tuning their claims to be more specific about what really matters here, than just saying vaguely about “transfer”.

---

> ### Author Response · Authors · 2020-11-20
> **Response to AnonReviewer2**
>
> We thank the reviewer for highlighting the insights that we would like to show in this paper and the appreciation of the significance of introducing NAS to the unrolling field.
>
>
> Q1: Weighted version of Figure 2.
>
> A1: It is a good point to give more weights to better architectures when calculating the average architecture. However, we found the top architectures perform almost equivalently with small variations in NMSE. Specifically, the top-1 architecture has -54.28 dB NMSE, the 50th has -54.21 dB and even the 100th has -54.19 dB, given the NMSEs of the whole search space are ranging from -46.0 to -54.3 dB. On this observation, we think taking the average architecture without weighting is a proper approach here.
>
> To further demonstrate the stability of our averaged architecture, we recalculate an averaged architecture from top-30 models, and compute its connection differences with the default top-50 average one. The difference map is visualized in the Appendix A.5, highlighted in red in the updated version of the paper. Clearly, they are highly aligned and echo our observations that those top architectures already have high agreement on connections to be activated.
>
>
> Q2: LWA version of LFISTA.
>
> A2: The LFISTA model we used in the paper already follows the same LWA parameterization. We mentioned this in the “Baselines” paragraph right before Section 3.2. We picked the LWA parameterization for fair comparison, in terms of the parameter complexity, because MM-based LFISTA contains one more weight matrix to train.
>
> We also train MM-based LFISTA as suggested. We find that in most settings, MM-based LFISTA is better than LWA-based LFISTA but still inferior to the searched architecture. For example, for noiseless, noisy and non-exactly sparse settings, the NMSE results of LWA LFISTA/MM LFISTA/Searched are (1) noiseless: -47.3/-53.2/-54.2; (2) noisy (SNR=20dB): -16.9/-17.7/-18.7; (3) non-exactly sparse: -26.3/-27.1/-29.1. However, MM LFISTA suffers when the noise type is mismatched, i.e. exp [h] in Table 1, where MM LFISTA has NMSE=-3.8dB for Gassian to S&P noise transfer while LWA LFISTA has NMSE=-9.2.
>
>
> Q3: Transferring the LISTA pattern to D-LADMM.
>
> A3: Thanks for all the terrific suggestions. We have conducted all your suggested experiments and reported results below. We (1) add dense connections to the latter half layers in D-LADMM (77 extra connections in total); (2) add 42 random connections (the same as the transferred LISTA pattern) to D-LADMM. All models are trained and evaluated in three representative settings as in the paper. For random connected models, we sample 5 architectures and report the average NMSE. The results are as follows.
>
>
> | Settings    | D-LADMM |  Dense Latter Half  |    Random   | Searched |
> |---------------|:----------:|:----------:|:----------:|:----------:|
> | Noiseless  |  -54.2   |  -55.3 | -55.04 | -55.6 |
> | Noisy (SNR=20dB)  | -18.2  |  -18.9  |  -18.92  |  -19.1 |
> | Limited data (10,240)  |  -24.8  |  -29.8  |  -25.56  |  -33.5  |
>
>
> We can see from the results that
> * Dense Latter Half does not outperform the transferred pattern, and especially degrades the performance when the training data becomes limited. It seems some earlier-layer extra connectivity might help the trainability of the early layer in those cases. That indicates that our learned connectivity pattern is more subtle than naively “denser for latter”.
> * For “adding random connections, but with the same total connected percentage”, those are in general on par with the Dense Latter Half, but even worse on limited data. Those show that although appropriate connection percentage is a useful factor, it does not constitute the major value of our searched specific connectivity .
>
> The extra experiments endorse our claim that the learned connectivity pattern on LISTA indeed conveys useful, nontrivial and transferable information that can even help ADMM.

---

> > ### Comment · AnonReviewer2 · 2020-11-21
> > **Thanks for the clarification. Confirm my strong support.**
> >
> > Title: Thanks for the clarification. Confirm my strong support.
> >
> >
> > Thank you for clarifying my questions and for providing the new figure/table. Great job!
> >
> > A3 is particularly interesting – that neither Dense Late Half nor random sparse seems to be a reasonable pattern. In comparison, the searched sparse connectivity transfers better in all cases. I’m very impressed and hope the authors continue investigating why it works so well, in their future work.
> >
> > The current work, IMHO, has clearly surpassed the bar of ICLR. I think it deserves a strong acceptance. The authors made an exceptionally comprehensive study for LISTA structures (rare for the unrolling field). The study is novel, in both the way it was designed & conducted and its resultant observations & insights. The performance gain by search is solid as examined in many settings.
> >
> > I strongly believe this study will be very impactful in the LISTA and model unrolling field. A lot of future readers will benefit.

---

### Decision · Program_Chairs · 2021-01-07
**Final Decision**

**Decision:**

Accept (Poster)

**Comment:**

The paper initially received mixed ratings, with one reviewer strongly supporting the paper given that the idea of combining unrolled algorithms and NAS is new and interesting, and one reviewer not convinced by the significance of the results. His/her main concern was the use of synthetic data only, which is not realistic. This was a legitimate concern as the performance of sparse estimation algorithms can change drastically when there is correlation in the design matrix. See for instance, the benchmarks in
F. Bach, R. Jenatton, J. Mairal and G. Obozinski. Optimization with Sparsity-Inducing Penalties.

The rebuttal addresses this concern in a satisfactory manner and the area chair is happy to recommend an accept.